# Unveiling the age and origin of biogenic aggregates produced by earthworm species with their NIRS fingerprint in a subalpine meadow of Central Pyrenees

Yamileth Domínguez-Haydar[1☯], Elena Velásquez[2☯], Anne Zangerlé[3¤], Patrick Lavelle[4☯], Silvia Gutiérrez-Eisman[5☯], Juan J. Jiménez[5☯‡]*

**1** Universidad del Atlántico, Barranquilla, Colombia, **2** Universidad Nacional de Colombia, Sede Palmira, Colombia, **3** Department of Ecology and Ecosystem Management, Technische Universität München, Freising, Germany, **4** Centre IRD Ile de France, UMR BIOEMCO 7618, Bondy Cedex, France, **5** Biodiversity Conservation and Ecosystem Restoration Department, Pyrenean Institute of Ecology (ARAID/IPE-CSIC), Jaca (Huesca), Spain

☯ These authors contributed equally to this work.
¤ Current address: Administration des services techniques de l'agriculture, Service agri-environnement, Luxembourg, Luxembourg
‡ This paper is dedicated to Juan G. Cobo, a dearest friend and colleague who suffered a tragic car accident that dramatically stopped his outstanding scientific performance in soil science studies.
* jjimenez@ipe.csic.es

**Data Availability Statement:** All relevant data are within the paper and its Supporting Information files.

## Abstract

In this study the near-infrared reflectance (NIR) spectra signals (750–2,500 nm) of soil samples was compared with the NIR signals of the biogenic aggregates produced in the lab by three earthworm species, i.e., *Aporrectodea rosea* (Savigny 1826), *Lumbricus friendi* Cognetti, 1904 and *Prosellodrilus pyrenaicus* (Cognetti, 1904) from subalpine meadows in the Central Pyrenees. NIR spectral signatures of biogenic aggregates, root-aggregates, and non-aggregated soil were obtained together with soil carbon (C), nitrogen (N), $NH_4^+$ and $NO_3^-$ determinations. The concentrations of C, N and C:N ratio in the three types of soil aggregates identified were not statistically significant (ANOVA, p>0.05) although non-macroaggregated soil had slightly higher C concentrations (66.3 g kg$^{-1}$ dry soil) than biogenic aggregates (earthworm- and root-aggregates, 64.9 and 63.5 g kg$^{-1}$ dry soil, respectively), while concentrations of $NH_4^+$ and $NO_3^-$ were highest in the root-attached aggregates (3.3 and 0.31 mg kg dry soil$^{-1}$). Total earthworm density and biomass in the sampled area was 137.6 ind. m$^{-2}$, and 55.2 g fresh weight m$^{-2}$, respectively. The biomass of aggregates attached to roots and non-macroaggregated soil was 122.3 and 134.8 g m$^{-2}$, respectively, while biomass of free (particulate) organic matter and invertebrate biogenic aggregates was 62.9 and 41.7 g m$^{-2}$, respectively. Multivariate analysis of NIR spectra signals of field aggregates separated root aggregates with high concentrations of $NH_4^+$ and $NO_3^-$ (41.5% of explained variance, axis I) from those biogenic aggregates, including root aggregates, with large concentrations of C and high C:N ratio (21.6% of total variability, axis II). Partial Least Square (PLS) regressions were used to compare NIR spectral signals of samples (casts and soil) and develop calibration equations relating these spectral data to those data obtained for

**Funding:** YDH was supported by a COLCIENCIAS grant, Colombia (Code: 1116-569-34827) for a stay at IPE-CSIC (Spain). EV, PL, AZ, SG did not receive specific funding. The funders had no role in study design, data collection and analysis, decision to publish, or preparation of the manuscript.

**Competing interests:** The authors have declared that no competing interests exist.

chemical variables in the lab. After a derivatization process, the NIR spectra of field aggregates were projected onto the PLS factorial plane of the NIR spectra from the lab incubation. The projection of the NIR spectral signals onto the PLSR models for C, N, $NH_4^+$ and $NO_3^-$ from casts produced and incubated in the lab allowed us to identify the species and the age of the field biogenic aggregates. Our hypothesis was to test whether field aggregates would match or be in the vicinity of the NIR signals that corresponded to a certain species and the age of the depositions produced in the lab. A NIRS biogenic background noise (BBN) is present in the soil as a result of earthworm activity. This study provides insights on how to analyse the role of these organisms in important ecological processes of soil macro-aggregation and associated organic matter dynamics by means of analyzing the BBN in the soil matrix.

## Introduction

The role of soil biota on aboveground and belowground processes is more and more recognized and precisely documented [1, 2]. Specifically, soil structure is one conspicuous product resulting from the activity of large invertebrates, the so-called bioturbators or soil ecosystem engineers (*sensu* [3]). Earthworms modify their environments, affecting the dynamics of soil organic matter and nutrient release and influencing the soil structure through the formation of biopores and aggregates [4]. The importance of such biological imprint remains less studied than purely physical processes [5]. The nature of these aggregates, their accumulation and distribution can affect soil critical processes including soil organic matter (SOM), water infiltration and soil C retention in aggregates [6–8].

One of the techniques that can be used to address these questions is near-infrared reflectance spectroscopy (NIRS, 750-2500nm). Itis a non-destructive, cost-effective, accurate and reliable method applied in different scientific disciplines including soil science [9–13]. Each biogenic structure in the soil has a specific NIRS signature, different from the surrounding soil [14]. This characteristic allows researchers to describe a typology of biogenic structures that relates to the organism which produced them. The NIRS technique has been widely used to assess the origin of aggregates such as biological, physical or root aggregates [15, 16], to identify the physical structures produced by soil ecosystem engineers [14, 17–19], and their age [20] and in soil quality studies [21–24].

Studies on the characterization of NIR spectral signals of soil aggregates contribute to quantify the functional role of soil ecosystem engineers and, particularly, earthworms [20–25]. Despite the scientific advances of those studies, the lifetime, degradation rate and the functional role of such structures in soil organic matter dynamics (SOM) is not well known, especially in field conditions. Zangerlé et al. [20] succeeded to discriminate the age of casts produced by the earthworm *Aporrectodea caliginosa* (Savigny, 1826) under laboratory conditions into three age classes (0 to 2, 3 to 30 and 45 to 90 days). The main difference in the NIR spectra signals was due to the loss of N in the biogenic structures until the signal camouflaged with the "bulk" soil. The prediction ability of NIRS to distinguish the origin (which species) and age (when was it produced) of soil aggregates in natural field conditions should be tested and validated [20]. As a result, a library of NIR spectral signals in existing ecosystems and under different field conditions should be generated to identify the origin of soil aggregates and its temporal degradation dynamics.

In another study Zangerlé et al. [25] succeeded to link NIR spectra of fresh casts, i.e. recently deposited on the soil surface, with their C contents. The question arises for what remains unknown, i.e., are we able to identify in field conditions and in the soil matrix the aggregates produced by a community of earthworms and when such structures were produced? The hypothesis tested in this study was that the origin of aggregates and their temporal dynamics can be distinguished in the soil matrix.

The main objective of our research was to unveil the contribution of earthworms in soil aggregation and soil organic matter (SOM) dynamics [26] in mountain soils. Secondary objectives were to (1) explore the capacity of NIRS to discriminate the age and origin (species) of a given soil aggregate (ped) under field conditions in subalpine pastures, and (2) obtain a NIRS library of earthworm casts (fresh and aging), and of biogenic soil aggregates, i.e., those of animal-origin and those attached to roots. It is expected that the combined NIRS-PLS methodology used in our study can be replicated with other earthworm communities and soil invertebrates producing biogenic structures.

## Materials and methods

### Study site

The study was conducted in the subalpine grasslands of the Ordesa and Monte Perdido National Park (OMPNP; 42˚36'N, 0˚01'E) in the Spanish Central Pyrenees. The climate is alpine, with annual average temperature and precipitation of 5 ˚C and 1,720 mm, respectively [27] (last 29 years at 2,200 m a.s.l.). The parent material is mainly calcareous substrates such as limestone, sandstone and flysch [28]. Extensive domestic grazing (goat, sheep and cattle) has been conducted during centuries in the area. Mountain grasslands of Central Pyrenees are facing physiognomic and physiologic changes related to socio-economic factors and climate trend. Grazing activities in the Pyrenees have shifted from sheep to cattle ranching and extensive husbandry has declined in the last decades [29, 30]. We thank permissions to access the research site by the mayor of the village of Fanlo (Huesca).

The sampling area shows evidence of livestock use (e.g., presence of cattle dung) at medium-stocking rates, and where gentler slopes allow cattle to move with less restrictions. The selected area is characterized by plant species within the Alliance *Bromion erecti* Koch, with *Festuca rubra* L., *Agrostis capillaris* L., *Trifolium pretense* L., *Lotus corniculatus* L. as dominant plant species [28, 31].

Soils where samples were taken are classified as Phaeozems by World Reference Base system [32]. They are characterized by high organic matter (OM) contents (11.9 and 5.8%), and pH ($H_2O$) ca. 7.1 and 7.3, and clay-loam and clayey texture for 0–20 and 20–50 cm soil depth, respectively [33].

### Soil macrofauna assessment and topsoil morphology analysis

In August 2014, earthworms and other soil macrofauna were hand-sorted in the field from five soil blocks of 25×25×20 $cm^3$ (ISO23611-5:2011). The soil cores were taken in a 400 $m^2$ quadrat, four in the corners and one in the center (Fig 1). Earthworms and macrofauna were taken to the laboratory, where they were washed and preserved in 4% formalin (10% of the commercial dilution) and 70% ethanol (macrofauna).

Earthworms were identified following available keys [34–36], and ascribed to the main ecological categories: epigeic, endogeic or anecic [35]. Soil macrofauna was ascribed to the main taxa groups.

The technique used to assess soil morphological analysis was derived from the "small volume" approach [15, 37]. In the area where soil macrofauna sampling was performed, nine soil

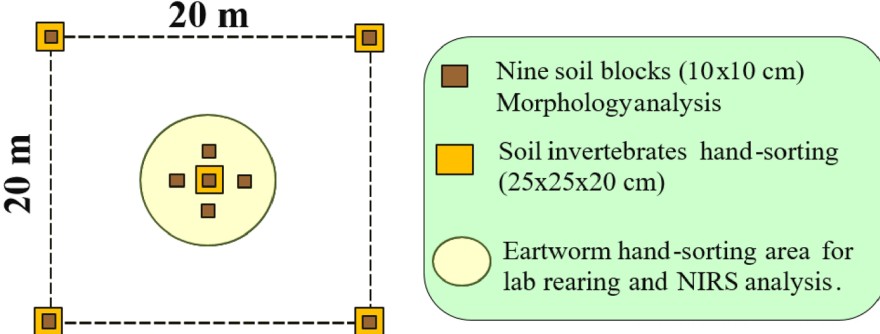

**Field sampling**

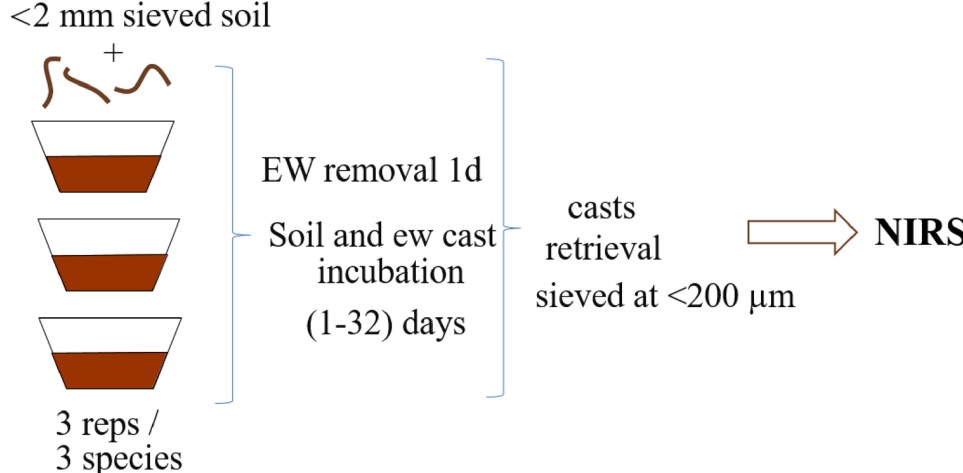

**Lab incubation**

**Fig 1. Experimental design and sampling strategy used in this study.**

cores of 10×10×10 cm$^3$ size were taken (Fig 1), carried to the lab. Soil macro-aggregates, i.e, larger than 0.250 mm where hand-sorted and separated into the following categories:

1. Invertebrate biogenic aggregates (BA): In general, these are produced by macroinvertebrates (mainly earthworms, Coleopteran larvae, and Diplopoda) and have generally rounded shapes and darker color than other soil aggregates. Earthworm casts generally comprise embedded round and concave structures corresponding to successive defecations of soil material into the galleries and macropores that they have previously created. Other macro-aggregates were included in this category whenever galleries or other structures associated with large invertebrates were visible on at least one side of the aggregate.

2. Root biogenic aggregates (RA): These are those aggregates attached to roots; if they have been produced by invertebrates, they are considered as "biogenic".

3. Physical aggregates (PA): These aggregates are mainly the result of physical processes (e.g., drying and rewetting of the soil).

How to effectively distinguish biogenic from physicogenic aggregates is key and it depends on the training of the observer/collector. When multivariate analyses are performed with this technique they are clearly separated in the factorial plane (see works by [15, 16, 18, 19, 22, 25, 38, 39].

The different types of aggregates (BA, RA, PA) were grounded after air-drying in the lab during several days. Each item separated during morphology analysis was weighed after drying to constant weight. Visible and near infrared (VNIR) spectra of the three types of aggregates and non-aggregated soil were obtained in a NIR spectrophotometer (ASD spec®) with a 350 to 2500 nm spectral range.

## Species rearing in the lab

One lab experiment was performed to obtain a reference library of NIR spectra of casts of different incubation times for all species. By the time of sampling, the species *Aporrectodea rosea* Savigny 1826, *Lumbricus friendi* Cognetti, 1904 and *Prosellodrilus pyrenaicus* (Cognetti, 1904) were collected in the field and reared in the lab. Mesocosm units were prepared with soil from the same area that was previously air dried and sieved at <2 mm. 200 g of air-dried soil and water added to field capacity (pF 2.5) were prepared one day before individuals were included in the mesocosms. A minimum of three to five individuals, depending on availability and in the same developmental stage, i.e, at least as subadults were put in each mesocosm for a total number of 21 (three replicates and 6 incubation dates). Average earthworm biomass in the mesocosms was 1.24 ± 0.11, 1.88 ± 0.18 and 3.0 ± 0.56 g.f.w. for *A. rosea*, *L. friendi* and *P. pyrenaicus*, respectively. Individuals were kept in mesocosm for one day and retrieved after casts were deposited in the soil volume. Soil with no casts and casts were incubated during 0, 1, 3, 7, 15, and 31 days, after which all casts were retrieved from the mesocosms, air-dried and sieved at 200 μm.

## NIRS library of aging casts

Samples were dried upon NIR spectra readings since water has a strong absorbance in the near-infrared spectra due to complicated hydrogen-bonding interactions at some specific wavelengths [40]. We used 10–15 g of each type of aggregate after grinding and sieving at 200 μm. Each sample (non-ingested soil and casts) was placed in a quartz cell and analyzed with a QualitySpec® spectrophotometer with light emission in the range 350–2500 nm, and reflected light measured at 1 nm intervals. The NIR spectra were obtained at 25 ˚C and 48% relative humidity. The spectra represented reflectance (R) of re-emitted light as a function of wavelength. Reflectance was converted to absorbance (A) using the equation: A = log (1/R). For each spectrum, the number of scans was set to be 10, which is higher than the recommended number of four to improve the signal to noise (S/N) ratio [41].

Concentrations of total C and N were determined by dry combustion in an Elementar Analyzer Vario Max CN (GmbH, Hanau, Germany) for all soil aggregates' types, after being grounded in a mortar and passed through a 250 $\mu$m sieve. Inorganic N concentrations in $NH_4^+$ and $NO_3^-$ forms were determined in triplicate with standard colorimetric methods after extraction with KCl 1M [42].

## Data analysis

**NIRS data normalisation and transformation.** A data matrix of NIR spectral signals can be highly heterogeneous and considerable variation of measured parameters may exist. Data pre-treatment is a prerequisite to reduce the number of non-informative variables (descriptors). First, we prepared a new data matrix with the absorbance values in the NIR spectra

region from 700 to 2,500 nm in 10 nm intervals. In our study we used two approaches to remove baseline shifts by preparing two data matrices with short- and long wavelengths, respectively:

1. one matrix with the absorbance data from 700 to 1,100 nm that was transformed to first derivative and,

2. a second matrix with absorbance data from 1,100 to 2,500 nm was transformed to second derivative using a $2^{nd}$ order polynomial Savitzky-Golay smoothing over 10 data (21 points) as the spectral pre-treatment method [43–45].

"Gap" size is an important issue during data transformation. It refers to the number of absorbance data either side of the spectrum point used in the derivative calculation. In other words, the length of the wavelength interval that separates the 2 segments to be averaged in the derivatization process. The segment size is decided by the observer, but in general ranges from 4 to 10 and even 20 wavelengths (nanometers). As expected, the gap size will affect the extent of smoothing of data, and it is dependent on the instrument used. A gap of 10 data points is higher than the minimum recommended (4) for calculation of derivative shifts [46], entailing averaging over 21 nm for second derivative. No further data pre-treatment was performed on the data matrix with derivative values.

**Multivariate analysis of NIR spectra.** Linear discriminant analysis (LDA) was used for data obtained in the lab, to obtain a typology of NIR signals for species and dates. The analysis was followed by a Montecarlo randomization procedure, 999 permutations) to search for significant differences among sites [47].

Partial least square regression (PLSR) was used to compare NIR spectral signals of samples. PLSR is one of the most commonly used techniques to analyze this type of data. It is a descriptor regression method for full data sets with a good capacity for estimating attributes based on the spectral behavior of the soil [13]. The analysis yields a number of factors from which a model is tested with new data to check the accuracy of the prediction. In our study we tested the models from the PLSR with the data of the soil variables quantified in the aggregates: C, N, $NH_4^+$ and $NO_3^-$.

**PLSR model selection.** The number of partial least-square (PLS) factors is chosen to minimize the root mean square error (RMSE) in the cross validation. The PLSR-NIPALS algorithm was used as it produces smaller prediction errors than NIPALS algorithm, improving the accuracy of the analysis. The performance of models was evaluated by the root mean square error of the cross-validation (RMSECV), i.e. the model with the lowest RMSECV is chosen as the optimal subset of descriptors [48]. Shenk et al. [46] recommended that the standard error of cross validation (SECV) should be no more than 20% greater than the standard error of calibration (SEC).

The suitability of samples for inclusion in the calibration population was tested with the Kennard-Stone algorithm [49]. Finally, the Hotelling's $T^2$ statistic [50] was used to identify outliers in the projections of NIRS field aggregates into the factorial plane of the selected PLS model (p<0.05).

Differences in the concentration of C, N, $NH_4^+$ and $NO_3^-$, and C:N in the different types of aggregates were tested with one-way ANOVA (p<0.05).

All analyses, i.e. data transformation, reduction, derivatization, PLSR and full cross-validation were conducted with the software Unscrambler X 10.5 (Camo software). The R statistical computing package (version 3.3.2) was used for multivariate PCA and Monte Carlo randomization [51].

**Table 1. Average density (N m$^{-2}$) and biomass (g f.w. m$^{-2}$) of soil macroinvertebrates and earthworms in the subalpine pasture at the study site (Central Pyrenees).**

| | Density (N m$^{-2}$) | Biomass (g m$^{-2}$) |
|---|---|---|
| **Soil macroinvertebrates (no EWs)** | 681.6 ± 486.3 | 5.62 ± 3.83 |
| **Earthworms** | 137.6 ± 61.5 | 55.17 ± 22.5 |

n = 5 25x25x25 cm$^3$ soil blocks.

## Results

### Soil macrofauna abundance and earthworm species

At the time of sampling, soil macroinvertebrate density and biomass (mean ± S.D.) was 681.6 ± 486.3 ind. m$^{-2}$, and 5.46 ± 3.71 g f.w. m$^{-2}$, respectively (Table 1, S1 Table, Supporting information). Total earthworm density and biomass was 137.6 ± 61.5 ind. m$^{-2}$, and 55.17 ± 22.5 g f.w. m$^{-2}$, respectively (Table 2). Five species were found and identified in the area where soil monoliths for NIR signatures were taken: *Aporrectodea rosea*, *A. caliginosa*, *L. friendi*, *Octolasion lacteum* Örley, 1881and *P. pyrenaicus*, with varying abundances (Table 2).

The biomass of aggregates attached to roots and non-macroaggregated soil was 122.3 and 134.8 g m$^{-2}$, respectively, while biomass of free (particulate) organic matter and invertebrate biogenic aggregates was 62.9 and 41.7 g m$^{-2}$, respectively (Table 3).

When these data were extrapolated to kg per meter square, the activity of earthworms resulted in 4.2 kg dry weight m$^{-2}$ of biogenic aggregates, and 12.2 kg dry weight m$^{-2}$ of aggregates attached to roots in the first 10 cm (Table 3). The amount of biogenic structures thus represented 45.4% of the total found in the sampled area, and 17.4% for free organic matter and 37.3% of bulk soil.

The concentrations of C, N and C:N ratio in the three types of soil aggregates identified (Table 4) did not differ statistically (ANOVA, p>0.05) although non-macroaggregated soil had slightly higher values than biogenic aggregates (earthworm- and root-aggregates), while concentrations of $NH_4^+$ and $NO_3^-$ were highest in the root-attached aggregates.

### Lab experimentation

**NIR spectra of casts produced and incubated in the lab.** When linear discriminant analysis was performed only with NIR spectra signals of casts produced by three species and

**Table 2. Density (N m$^{-2}$) and biomass (g m$^{-2}$) of earthworm species found in each soil core (25x25 x 20 cm$^3$) in the subalpine pasture at "Ordesa and Monte Perdido" National Park.**

| Core | Species | Ecological category | Adults | Subadults | Inmatures | Density | Biomass |
|---|---|---|---|---|---|---|---|
| 1 | *L. friendi* | Epi-endogeic | 0 | 0 | 3 | 48 | 1.0 |
| | *O. lacteum* | Endogeic | 0 | 0 | 3 | 48 | 0.62 |
| 2 | *A. caliginosa* | Epi-endogeic | 0 | 1 | 0 | 16 | 5.84 |
| | *P. pyrenaicus* | Endogeic | 2 | 3 | 0 | 80 | 8.5 |
| 3 | *A. rosea* | Endogeic | 1 | 0 | 0 | 16 | 3.07 |
| | *P. pyrenaicus* | Endogeic | 1 | 1 | 3 | 80 | 5.58 |
| 4 | *L. friendi* | Epi-endogic | 1 | 0 | 0 | 16 | 21.22 |
| | *A. rosea* | Endogeic | 2 | 0 | 1 | 48 | 2.29 |
| 5 | *P. pyrenaicus* | Endogeic | 2 | 3 | 6 | 176 | 5.6 |
| | **Total** | | **9** | **8** | **16 Mean** | **105.6** | **10.74** |
| | | | | | **S.D.** | **50.6** | **6.27** |

**Table 3. Weight of the different elements from the 9 soil blocks used for micromorphology analysis.** Data correspond to average values.

| Soil block # | Weight of soil macro-aggregates (g dry soil $cm^{-2}$) | | | |
|---|---|---|---|---|
| | Biogenically produced by invertebrate | Attached to roots | Particulate organic matter | Not identified asmacro-aggregate |
| 1 | 10.3 | 175 | 99 | 120 |
| 2 | 18.1 | 161.7 | 61.6 | 68 |
| 3 | 7.7 | 9.1 | 67 | 171 |
| 4 | 0.9 | 38 | 73.2 | 251 |
| 5 | 20.7 | 106 | 49.2 | 126 |
| 6 | 11 | 92 | 74.5 | 73 |
| 7 | 243 | 105 | 29 | 200 |
| 8 | 38.8 | 264 | 42.5 | 103.2 |
| 9 | 24.6 | 150.5 | 70.3 | 101 |
| Total | 375.1 | 1,101.3 | 566.3 | 1,213.2 |
| Mean ± S.D. | 41.7 ± 76.3 | 122.3 ± 76.3 | 62.92 ± 20.5 | 134.8 ± 61.02 |

considering all dates (Fig 2a), the amount of explained variance by the first three axes was 61.3% after Montecarlo randomization (p<0.001). The wavelength spectrum selected was based on the visualization of the second derivative transformation, looking at the areas where peaks and valleys appeared (not shown). The selected wavelengths ranged from 1,350 to 2,260 nm. The barycenter of species and cast age were separated from the control soil. The age of casts had more influence on NIR spectra than species. The first axis (25.3% of total variation) separated 2-days old casts of *A.rosea* from the rest of incubation dates (Fig 2a). The second axis contributed 20.4% to total explained variability and separated the 32-days old casts of *L. friendi* and *P. pyrenaicus* from the most recent casts (1–4 days old) of the different earthworm species. The ordination was significant after Montecarlo randomization (p<0.001). When using axes III and IV (24% of total variability explained) 8-days old casts were opposed to 16-d old casts for the three species along axis III (Fig 2b). Age of the cast rather than species identity was the most important factor that determined the grouping of casts in the factorial plane, i.e., incubation time drives changes between the most recent casts and the older casts.

The LDA with the NIR spectral signals of incubated casts produced by each species showed a clear separation of the cast incubation time (Fig 3a–3f):

- *Aporrectodea rosea*, the first two axes explained 73.4% of total variance (68.3% after Monte Carlo randomization). The older incubated casts were related to high concentrations of C, N and $NO_3^-$ (Fig 3a), while 2-days old casts were related to high C:N ratio (Fig 3b).

**Table 4. Concentrations of total C, N and inorganic N (N-NO$_3$ and N-NH$_4$) in the three types of soil aggregates identified.**

| Aggregate | | Carbon[1] | Nitrogen[1] | C:N | N-NH$_4$[§] | N-NO$_3$[§] | n |
|---|---|---|---|---|---|---|---|
| **Biogenic** | Mean | 64.9 | 6.8 | 9.8 | 2.6 | 0.16 | 17 |
| | SD | 9.2 | 0.8 | 0.4 | 1.6 | 0.5 | |
| **Root-attached** | Mean | 63.5 | 6.9 | 9.7 | 3.3 | 0.31 | 17 |
| | SD | 9.1 | 0.6 | 0.5 | 3.2 | 0.8 | |
| **Non-aggregated** | Mean | 66.3 | 7.0 | 9.8 | 3.4 | 0.06 | 18 |
| | SD | 7.8 | 0.5 | 0.3 | 1.9 | 0.16 | |

[1] g kg dry soil$^{-1}$.

[§] mg kg dry soil$^{-1}$.

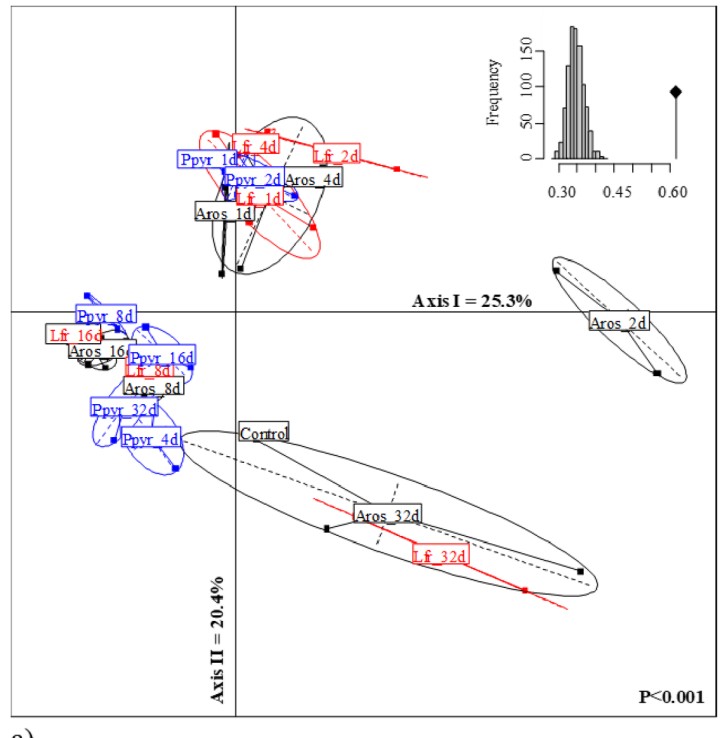

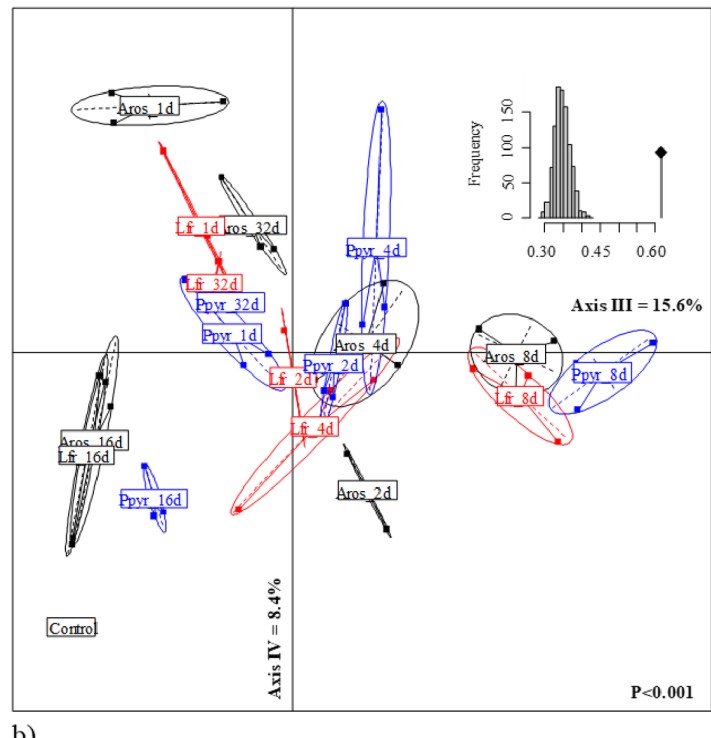

**Fig 2. LDA of the NIR spectra signals of incubated casts in the mesocosm units during 32 days.** a) factorial plane of the first two axes; b) factorial ordination by axes III and IV; Aros = *A. rosea*; Lfr = *L. friendi*; Ppyr = *P. pyrenaicus*; NA = Non-aggregated soil. CT = Control soil.

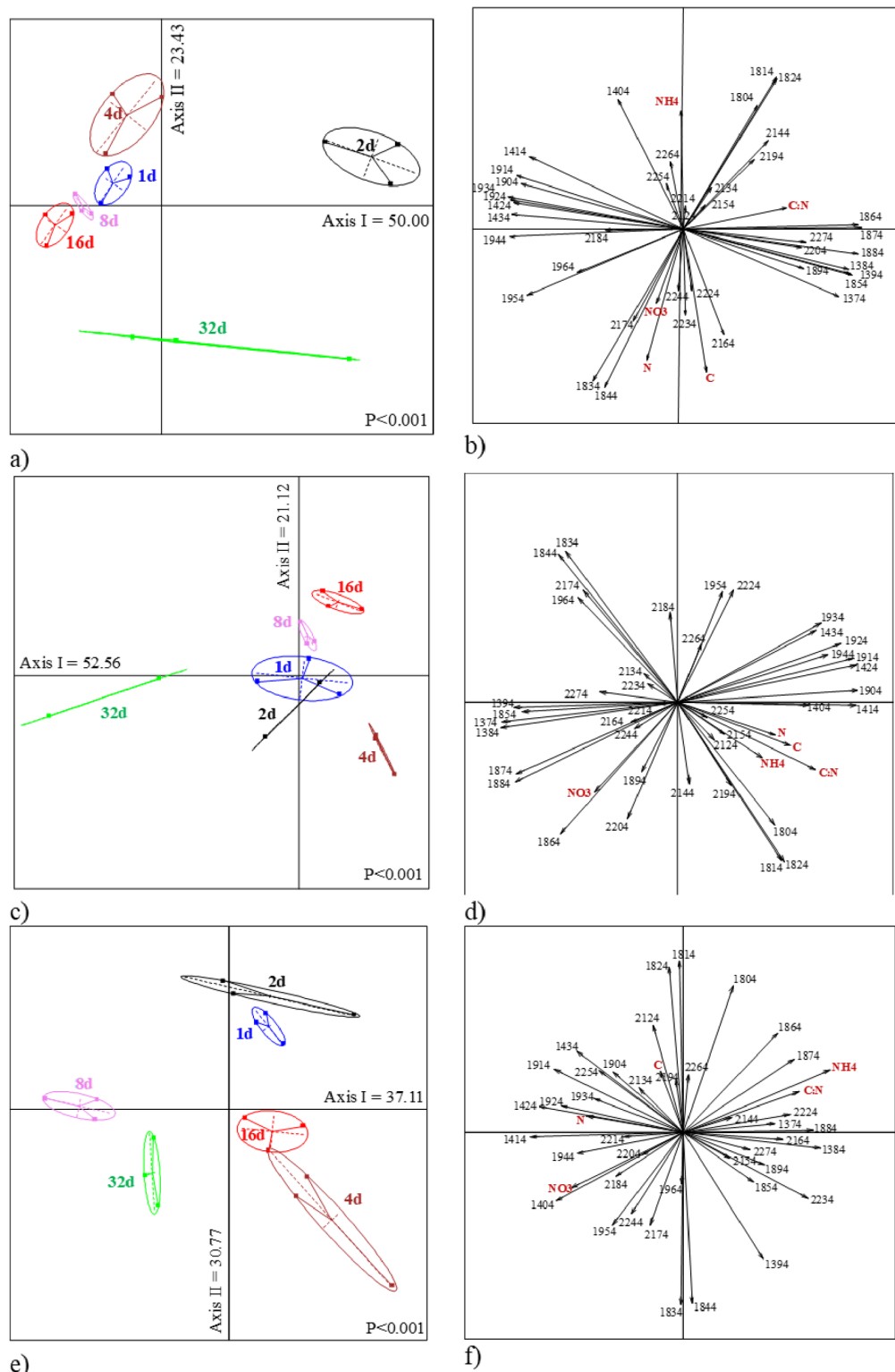

**Fig 3. LDA of the NIR spectra of casts produced by the three species under study and incubated in the lab during 32 days and ordination of variables (NIR wavelengths and measured variables).** Earthworm species are *A. rosea* (a, b), *L. friendi* (c, d) and *P. pyrenaicus* (e, f). All ordinations were significant after Monte Carlo simulation (999 sim).

- *Lumbricus friendi*, axes I and II explained 52.6 and 21.2% of total variance, respectively (65.1% after randomization), and similar to *A. rosea* 32-days old casts were clearly separated from the other incubation times and was related to high values of $NO_3^-$ concentrations (Fig 3c and 3d). In these two species, 4-days old casts were correlated to high concentrations of $NH_4^+$.

- *Prosellodrilus pyrenaicus*, the first two axes explained 67.9% of total variability (observed value equaled 62.5% after randomization process). Again, for this species, 32-days old casts were related to high concentrations of $NO_3^-$ (Fig 3e and 3f).

The total amount of variance explained by axes III and IV was 24.5% (14.0 and 10.5%), 21.0% (14.2 and 6.8%), and 28.6% (18.0 and 10.6%) for *A. rosea*, *L. friendi* and *P. pyrenaicus*, respectively (Fig 4a–4f). The ordination of incubation dates and variables in the factorial plane of axes III and IV showed that C:N ratio was related to 32-days incubated casts of *A. rosea*, and that $NH_4^+$ was for 4-days old casts of *L. friendi* and *P. pyrenaicus*, while $NO_3^-$ was related with 16-days old casts of *P. pyrenaicus* (Fig 4a–4f).

These multivariate ordinations constitute specific NIR spectra reference library of ageing casts produced and incubated in the lab for the three earthworm species. The next step was to project the NIR spectral signals of the distinct aggregates collected in the field to examine whether their location in the factorial plane could be matched with certain clusters of ageing casts (see section "Relating NIR spectral signals of field soil aggregates with lab-incubated earthworm casts").

**Calibration model performance and PLSR.** The results of the NIRS calibration and validation for lab incubated casts is indicated in S2 and S3 Tables and S1–S6 Figs (Supporting information). PLSR calibration model statistics varied among species and variables, i.e., $R^2_{cal}$ varied from 0.477 to 0.997 and RMSEC from 0.006 to 5.62 for 1$^{st}$ derivative NIR data (S2 Table, Supporting information). For 2$^{nd}$ derivative NIR data, $R^2_{cal}$ varied from 0.516 to 0.999 and RMSEC from 0.002 to 5.01 (S3 Table, Supporting information).

The derivative treatment of absorbance values influenced the calibration process. The gap used of ten points (21 in total) when using first derivative $R^2_{cal}$ was 0.86 and RMSECV = 0.43 for 6 factors. When using second derivative and the same gap size of 21 points $R^2_{cal}$ and RMSECV was 0.92 and 0.37. In our study $R^2_{cal}$, RMSECV and the number of factors from the PLS varied (S2 and S3 Tables). When $R^2_{cal}$ was above 0.9 the number of factors ranged between 4–7 when using 1$^{st}$ derivative short wave NIR data (700–1,100 nm). On the contrary less number of factors were obtained when using second derivative NIR data, i.e., from 3 to 6 factors for $R^2_{cal}$ above 0.9 (S3 Table). In other words, the precision of the validation increases when we use second derivative NIRS data. In general, these values are within the range indicated by Guthrie 2005 when analyzing calibration model performance in a study with total soluble solids, i.e. $R^2_{cal}$ = 0.92, and RMSECV = 0.37 for 7 factors (latent variables).

The best models when using short wave NIR (700–1,100 nm) for *A. rosea* were obtained for C ($R^2_{cal}$ = 0.964) and $NH_4^+$ concentrations ($R^2_{cal.}$ = 0.996). The SECV was higher than 20% greater than the SEC for all variables (S2 Table, S1 Fig). High $R^2_{cal}$ were obtained for the five parameters analysed in *L. friendi* (S2 Table, S2 Fig), but the SECV was in all cases 20% higher than SEC. In *P. pyrenaicus*, only for C:N ratio and $NH_4^+$ concentration $R^2_{cal}$ was above 0.99. The SECV was higher than 20% of the SEC (S2 Table, S3 Fig).

When analyzing long wavelength NIR spectra (1,100–2,500 nm) for *A. rosea* the best models were obtained for C ($R^2_{cal}$ = 0.998) and $NH_4^+$ concentration ($R^2_{cal}$ = 0.99); again, the SECV was higher than 20% of the SEC for all variables (S3 Table, S4 Fig). In *L. friendi* high $R^2_{cal}$ were

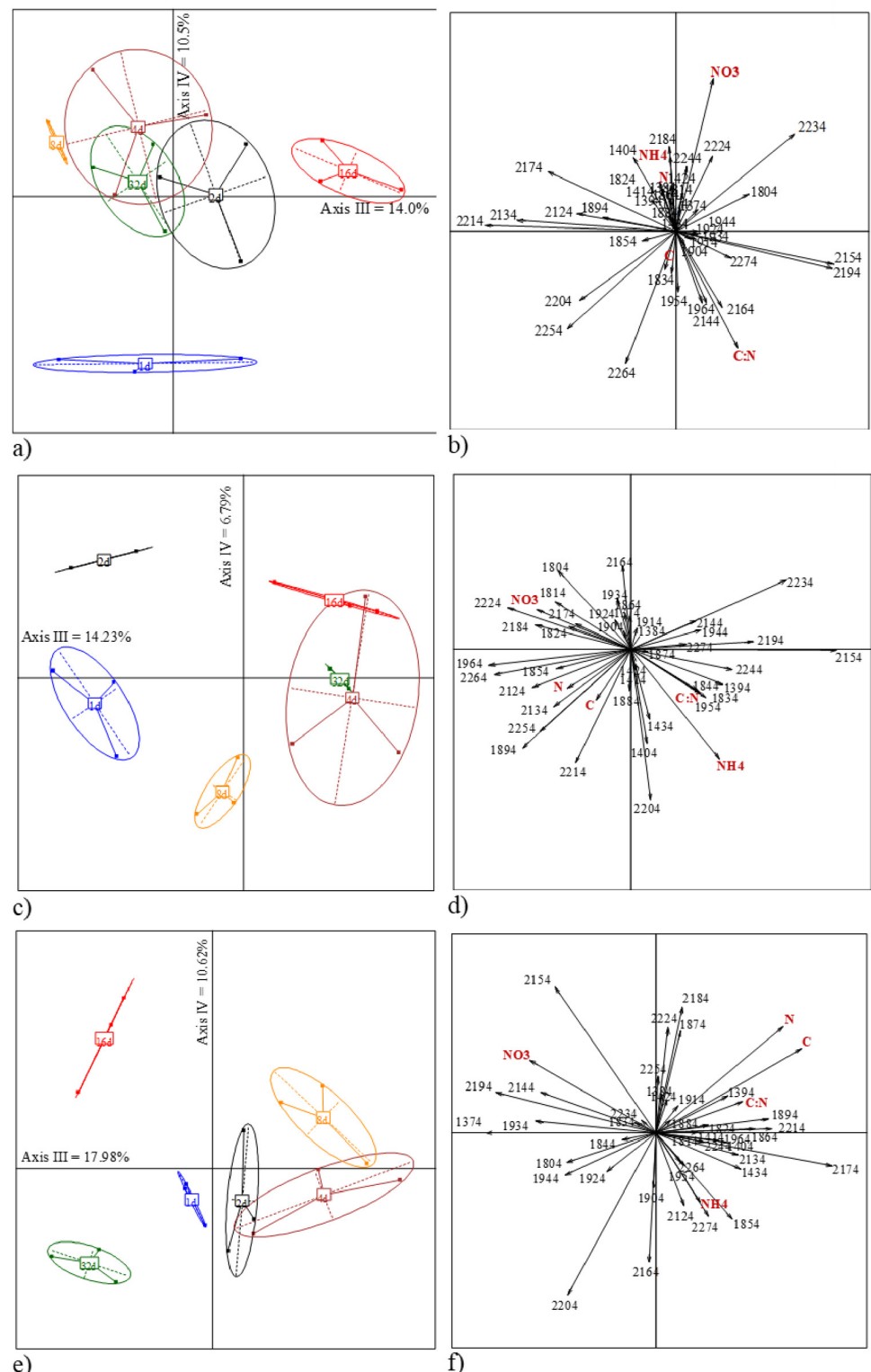

**Fig 4. Biplot of the NIR spectra along axes III and IV of the PCA for the three species under study and ordination of variables (NIR wavelengths and measured variables) for all the casts incubated in the lab during 32 days.** *A. rosea* (a, b), *L. friendi* (c, d) and *P. pyrenaicus* (e, f).

obtained for C:N ratio and $NH_4^+$ concentration (S3 Table, S5 Fig), but the SECV was in all cases 20% higher than SEC. Finally, in *P. pyrenaicus* high $R^2_{cal}$ values were obtained for the five parameters analyzed (S3 Table, S6 Fig), and the SECV was higher than 20% of the SEC.

## Linking NIR spectra of field soil aggregates with lab-incubated earthworm casts

The projection of the NIR spectral signals onto the PLSR models for C, N, $NH_4^+$ and $NO_3^-$ from casts produced and incubated in the lab allowed us to identify the species and the age of some of the field biogenic aggregates (Figs 5–8). The PLSR models were obtained from selection of wavelengths after visualization of the spectrogram, i.e, 1$^{st}$ and 2$^{nd}$ derivative transformation (Savitzky-Golay) for short- and long-wave NIR spectral signals, respectively. The accuracy of results in the PLSR biplots was checked with the Hotelling's $T^2$ ellipse ($p < 0.05$).

In all depicted biplots field biogenic aggregates were near the vicinity of species-specific NIR spectral signals of casts of different age. In other words, the biogenic aggregates were close to the cloud of points (NIR background noise) of the ageing NIR spectral signals of incubated casts of the three species. The pattern was more evident for PLSR ordination of long-wave NIR spectral signals (1,100–2,500 nm) for all variables.

**Carbon.** When using short-wave NIR spectral signals, the first and second axes explained 98% and 1% of variance, with the PLSR explaining 20 and 42%, respectively (Fig 5a). In the factorial plane, BA3 was near the NIR signals of 16-d old casts produced by *P. pyrenaicus*. Less clear pattern appeared for BA4, BA6 and BA8 were in the space defined by 1-day old casts of *L. friendi* and *P. pyrenaicus* (Fig 5a).

When using long-wave NIR spectral signals for the same variable, the first and second axes explained 67% and 10% of variance, with the PLSR explaining 21 and 9%, respectively (Fig 5b). In the factorial plane, BA1 was in an area with NIR signals of casts older than 4 days, i.e., close to *P. pyrenaicus* 4-d, *A. rosea* 16-d and *P. pyrenaicus* 32-day old casts. NIR signal of BA3 was close to *A. rosea* 4-day old casts, while BA5 and BA8 were close to *L. friendi* and *A. rosea* 1-day old casts, respectively (Fig 5b). BA 4 to BA9 NIR spectra were in the area of the factorial plane where NIR signals of casts were less than 4-days old. Regarding C contents, NIR spectra signal of older casts were quite similar independently of the species which produced the casts (Fig 5a and 5b).

**Nitrogen.** When using short-wave NIR spectral signals, the first and second axes explained 98% and 1% of variance, with the PLSR explaining 20 and 43%, respectively (Fig 6a). In the factorial plane, NIR signals of BA3 and BA4 were near the NIR signals of 16-d old casts produced by *P. pyrenaicus*. BA6 and BA8 were in the space defined by 1-day old casts of *L. friendi* and *P. pyrenaicus* (Fig 6a).

When using long-wave NIR spectra signals, the first two axes explained 61 and 17% of total variance, and the PLSR explained 22 and 10% for axis 1 and 2, respectively (Fig 6b). In the factorial plane NIR spectra of BA1 and BA2 were in and area close to *P. pyrenaicus* 4-d old casts, and *L. friendi* 16-days old casts, respectively. NIR signals of BA3 were close to those of *A. rosea* 4-day old casts, while BA4, BA5, BA6 NIR signals were close to those represented by *L. friendi* 1-day old casts, and BA8 and BA9 to *A. rosea* and *P. pyrenaicus* 1-day old casts, respectively (Fig 6b).

**Ammonium.** When using short-wave NIR spectral signals, the first and second axes explained 98% and 1% of variance, with the PLSR explaining 22 and 45%, respectively (Fig 7a). In the factorial plane, NIR signals of BA3 were near the NIR signals of *A. rosea* 4-d old casts. NIR signals of BA4, BA6 and BA8 were in the space defined by 1-day old casts of *L. friendi* and *P. pyrenaicus* (Fig 7a).

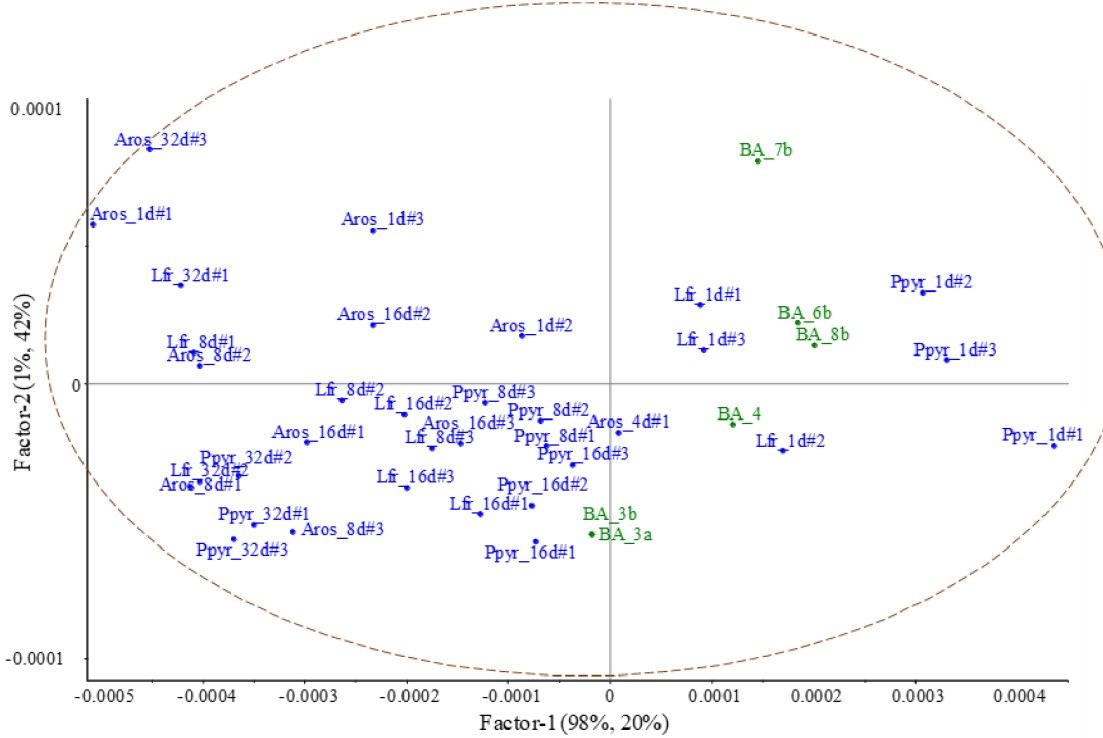

a)

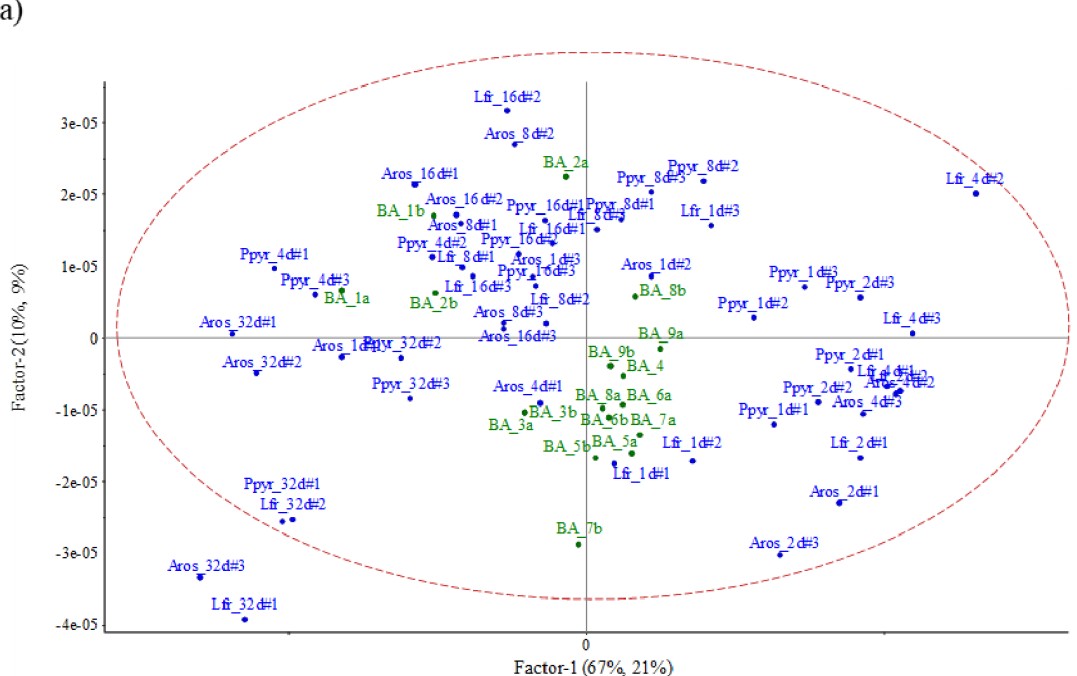

b)

**Fig 5. Projection of short-wave (a) and long-wave (b) NIR signal of test samples (field aggregates) onto the PLSR model for C concentration in casts from lab experiment (all dates).** Hotelling's $T^2$ confidence limit ($p < 0.05$) is represented by an ellipse. BA = field biogenic aggregate; Aros = *A. rosea*; Lfr = *L. friendi*; Ppyr = *P. pyrenaicus*.

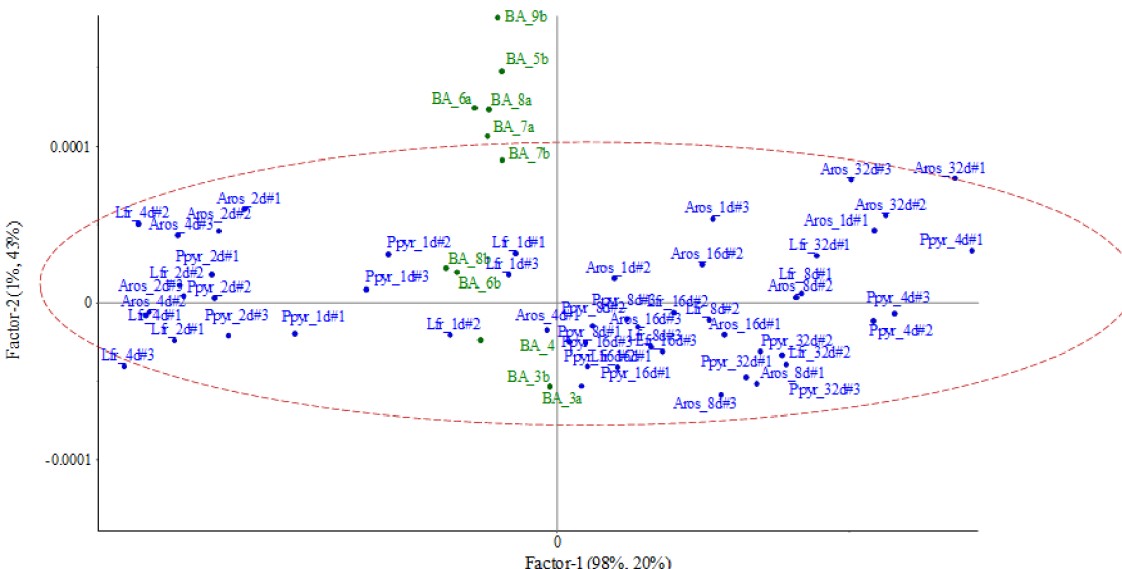

a)

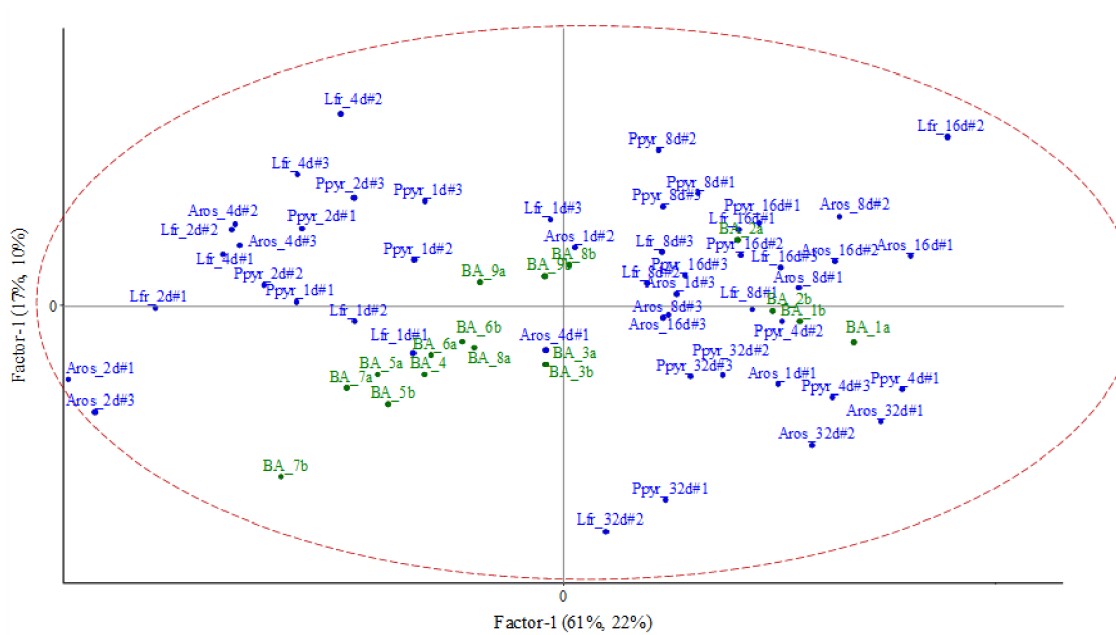

b)

**Fig 6. Projection of short-wave (a) and long-wave (b) NIR signal of test samples (field aggregates) onto the PLSR model for N concentration in casts from lab experiment (all dates).** Hotelling's $T^2$ confidence limit (p<0.05) is represented by an ellipse. BA = field biogenic aggregate; Aros = *A. rosea*; Lfr = *L. friendi*; Ppyr = *P. pyrenaicus*.

When using long-wave NIR spectral signals, the first axis explained 71% of variance and the second axis 2%, with the PLSR explaining 25 and 3%, respectively (Fig 7b). BA2 was close to *L. friendi* 8-day old casts, BA5 to *L. friendi* 1-day old casts, BA3 and BA8 to *P. pyrenaicus* 8-days old casts (Fig 7b).

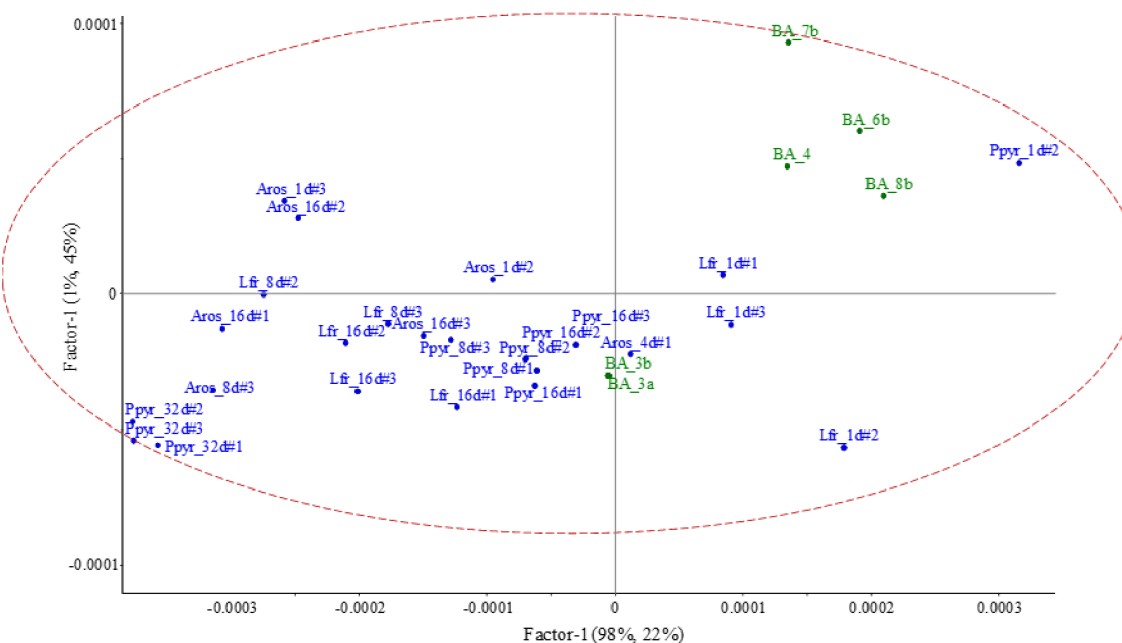

a)

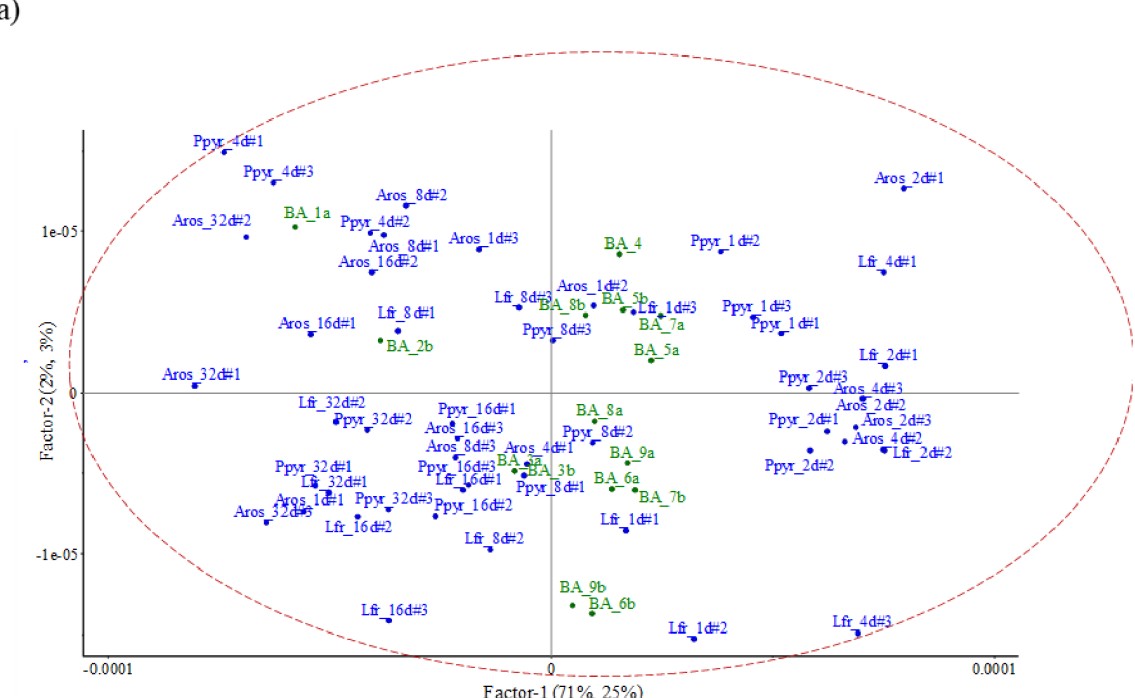

b)

**Fig 7. Projection of short-wave (a) and long-wave (b) NIR signal of test samples (field aggregates) onto the PLSR model for NH4 concentration in casts from lab experiment (all dates).** Hotelling's $T^2$ confidence limit ($p<0.05$) is represented by an ellipse. BA = field biogenic aggregate; Aros = *A. rosea*; Lfr = *L. friendi*; Ppyr = *P. pyrenaicus*.

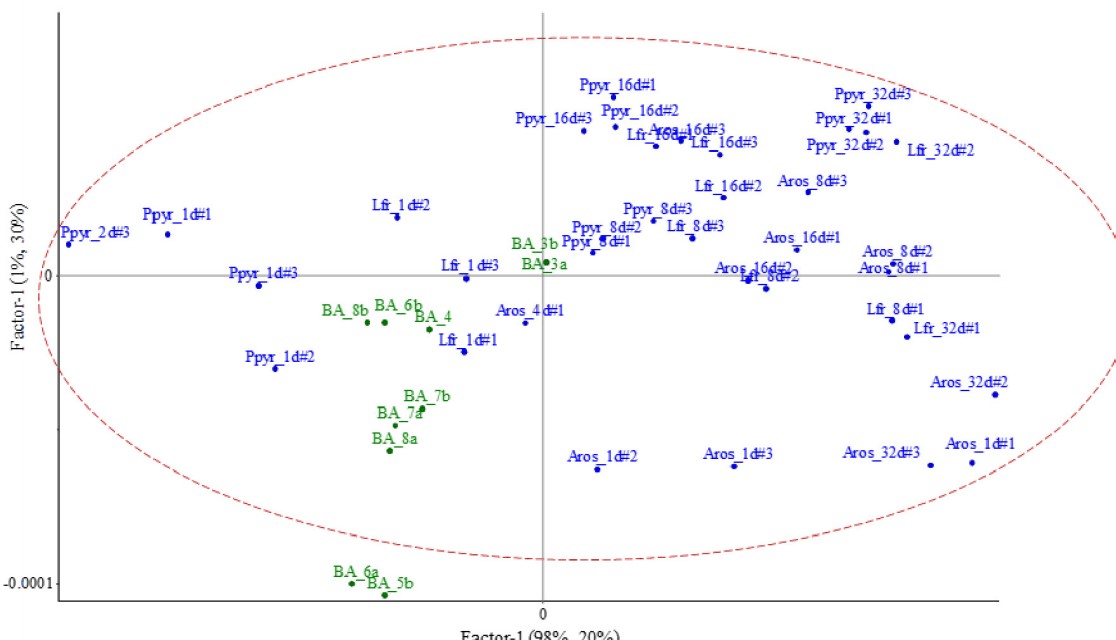

a)

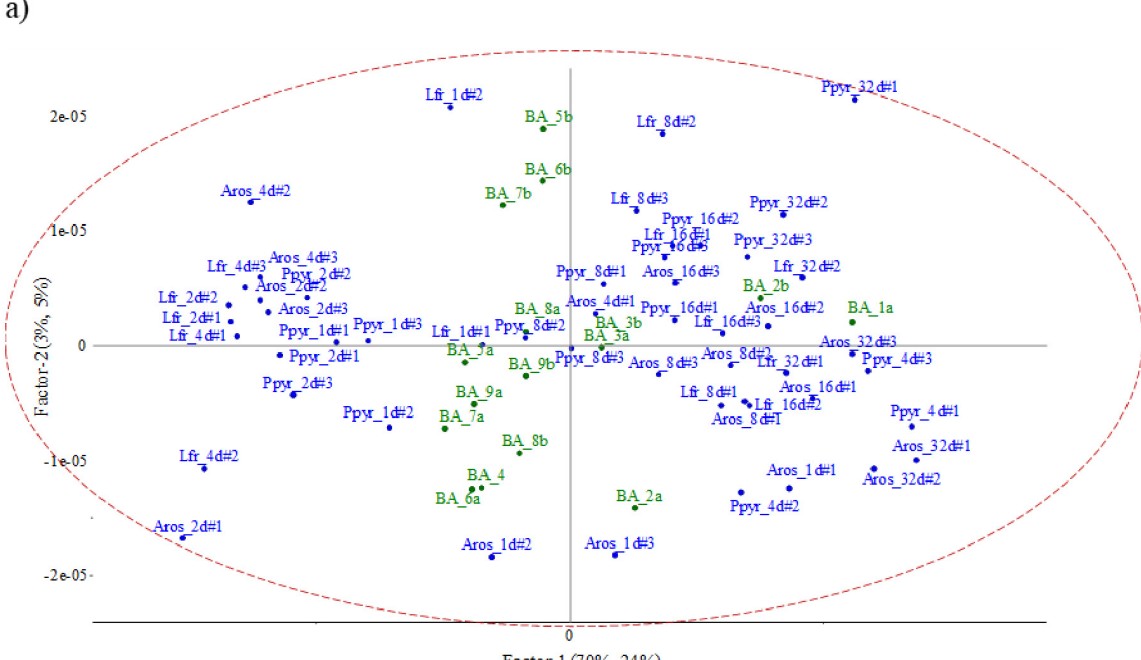

b)

**Fig 8. Projection of short-wave (a) and long-wave (b) NIR signal of test samples (field aggregates) onto the PLSR model for NO3 concentration in casts from lab experiment (all dates).** Hotelling's $T^2$ confidence limit ($p<0.05$) is represented by an ellipse. BA = field biogenic aggregate; Aros = *A. rosea*; Lfr = *L. friendi*; Ppyr = *P. pyrenaicus*.

**Nitrate.** When using short-wave NIR spectral signals, the first and second axes explained 98% and 1% of variance, with the PLSR explaining 20 and 30%, respectively (Fig 8a). In the factorial plane, NIR signals of BA3 were near the NIR signals of casts of *A. rosea* 4-d old casts and of casts of *P. pyrenaicus* 8-d old. NIR signals of BA4, BA6, BA7 and BA8 were in the space defined by 1-day old casts of *L. friendi* and *P. pyrenaicus* (Fig 8a).

Finally, for $NO_3^-$, when using long-wave NIR spectral signals, the first two axes explained 70 and 3% of total variance, and the PLSR explained 24 and 5% for axes 1 and 2, respectively (Fig 8b). In the ordination plot the biogenic aggregates were farther separated from the NIR spectral signals of lab incubated casts than with the rest of variables. Only NIR spectra of BA3 and BA8 were close to *P. pyrenaicus* 8-days old casts, and BA5 to *L. friendi* 1-day old casts (Fig 8b). The NIR signals of BA1 and BA2 were located near the NIR signals of older casts, i.e. 16–32 days-old of the three species. No clear pattern was observed for the rest of BA, which were in the area of 1–2 days-old casts.

## Discussion

Comparison of our results with those obtained in other studies is not possible as the approach used in this study, i.e., linking field aggregates of the soil matrix, not only recently deposited or fresh casts, to NIR signals was not assessed in previous works. The biogenic structures produced by earthworms can be used to establish a typology of structures linked to functionality in the ecosystem where they are found [14–16]. In other studies, NIR signals were related with the age of the biogenic aggregates to assess their origin (root, earthworm or mixed structures [18], and their temporal dynamics [20]. Although some limitations have been raised by several authors about the use of NIRS [52], those may be well linked to specific local heterogeneity processes [53]. Soil aggregation dynamics can be very fast until the NIR spectra signal resembles that from the bulk soil [20]) so they are not distinguishable.

Higher concentrations of C, N and the C:N ratio were observed in the non-macroaggregated soil than in the biogenic aggregates, while $NH_4^+$ and $NO_3^-$ concentrations were highest in the aggregates attached to roots (differences were not significant, ANOVA). These results could be explained by the presence of free particulate organic matter in the non-macroaggregated soil, while the concentrations of $NO_3^-$ and $NH_4^+$ in the biogenic root-attached aggregates would be the result of the microbial interactions in the rhizosphere. This hypothesis should be further tested. Root aggregates are also biogenic aggregates and thus confusion may arise occasionally when trying to distinguish roots- and invertebrates' biogenic aggregates, especially when roots grow in casts as observed in some studies [18].

### NIR assessment of field aggregates and casts incubated in the lab

In previous studies the NIR signatures of biogenic structures produced by different ecosystem engineers in the field were analysed [14, 15, 19, 20]. To our knowledge, only one field study assessed how well the NIR spectra signals of field aggregates matched up with those signals obtained from casts of earthworm species incubated in the lab [25]. These authors collected fresh casts from the field to compare and match their NIR signals with the casts produced by the same species in the lab. Our study, on the contrary, is the first one aiming at pairing the NIRS spectra fingerprints of ageing biogenic aggregates with the respective species that were likely responsible of forming those aggregates.

Comparisons with other studies are thus not convenient, and only comparisons about the relevance of NIRS as a tool to discriminate biogenic aggregates from other structures found in the soil are possible.

Altogether, the NIR spectra fingerprints were distinguished in the multivariate analyses (Figs 2–4). In general, the multivariate analyses performed on the NIR spectral discriminated more clearly the age of the biogenic aggregate of each species reared in the lab, although not as accurately ordered as in the work by Zangerlé et al. [20]. During ageing, there is an initial phase (0–1 day) where microbial biomass and $NH_4^+$ are high and later these two parameters decrease and an increase of $NO_3^-$ contents. In our study this is a striking result as each replicate was obtained from a different mesocosms that were kept in the same conditions in the lab and so the signals projected for each age were grouped together in the factorial plane.

## Pros and cons of the intended NIR assessment approach in field conditions

The usefulness of using NIRS to differentiate cast aggregation may be context-dependent [53]. In our study fresh casts of the three reared species had similar NIR signatures, i.e., there was no clear specificity of depositions according to the species that produced them. For each species reared in the lab clusters of aged casts were discerned in the multivariate analyses showed. The age of casts more strongly influenced on NIR spectra than the cast-forming species. The ordination of biplots in Figs 5–8 indicated that the first axis separated the biogenic structures by their age, while the second axis was more linked to the functional role of species. This could indicate that the same process is occurring in the gut of different species, i.e., that they have similar microbial populations to help them digest the different compounds of the resources they ingest in the soil [54].

However, despite the ability of NIRS to identify biogenic aggregates only three earthworm species were collected and reared in the lab, as two other species, i.e. *Octolasion lacteum* and *O. cyaneum* (Savigny, 1826) were not found at the time of sampling, so no casts were available for both species. This resulted in an incomplete picture of the processes driven by earthworm activity. Zangerlé et al. [20] were able to identify and classify casts produced by the endogeic earthworm *A. caliginosa* in the lab up to three months after these were produced. The NIR spectral signals were similar for casts ranging from 45 to 90 days old, and also from 3 to 30 days old. In our study, on the contrary, the ordination of NIR spectral signals of casts of different age were clearly separated from 1 to 21 days old (Fig 2a, 2c and 2e).

In our study the age of casts was most important than species identity. Not all BA retrieved from the soil cores were produced by the same species and in the same time lag. This resulted in observing similar age of casts but different species in the multifactorial plane of the PLSR models (Figs 4–7). BA of the same age but produced by different species were grouped in the ordination. Thus, as predicted by Zangerlé et al. [20, 25], we observed in these figures a concurrence of BA that were produced by different species, i.e., similar NIR spectra for casts of different species which participated in their building. This represents the biogenic background noise (BBN) of the NIR spectral signals of casts.

We thus assumed that the BA collected from the soil cores were likely produced by several species and that these had different ages at the time of sampling. In the process of analysis, we included all the BA into the same pool for NIR readings. As a result, the sample for NIR spectrophotometer is a mixture of biogenic structures produced by different species of varying age. Only when an individual is found in the soil core we suppose that some of the aggregates were produced by that specific species, but even in this case uncertainty prevails.

The field NIR spectral signals were grouped together in the vicinity of lab models and are not widely scattered in areas where no model projections exist in the factorial plane. Here we provided data that constitute a spectra baseline that can be used for further assessment of the functionality of the species found in the area. We aimed to increase knowledge on the importance of different types of aggregates found in the soil with a methodology that needs further

to be consolidated. Soils and the structures found therein and produced by the bioturbation of soil animals, are inherently non-linear systems, which may limit our ability to predict important soil ecological processes. A complete assessment of the BBN in further studies will result in a more precise soil C modelling [7, 8, 55].

Although NIRS offers an excellent opportunity for soil ecologists to escape the constraints of laboratory analyses, at the same time, this study involved an intensive amount of work both in the field and in the preparation of the mesocosms. The limitations were given by retrieving a sufficient number of specimens of all species present in the area under study. One caveat worth mentioning is related to the process of identifying and separating BA from the soil block. The expertise of field assistants, students and researchers is a must; otherwise, this imprecise selection of field aggregates may lead to undesirable mistakes in the NIR-PLS methodology. NIRS also provides the advantage to analyse other soil processes that lab analyses do not achieve, i.e., which species was responsible for the formation of a given aggregate and when it was produced.

Finally, how the biologically-produced aggregates in soils where several species coexist can be separated effectively if they are later grouped together to form a single sample for NIR spectrophotometer? By increasing our understanding of the processes that are affected by earthworm ecosystem engineering (bioturbation) we shall better identify and quantify specific functions that soil organisms fulfill in ecosystems.

## Conclusions

In this study we explored the feasibility of NIRS to identify the type, the "who-did-it and when" of soil biogenic aggregates or casts that were formed by several earthworm species that were present in a mountain meadow. We showed that NIRS was a suitable, powerful and rapid technique for such predictions, more so when second derivative NIR spectra data are used. The resulting models that were built with PLSR of NIR spectral signals showed the existence of an underlying mechanism that needs to be further investigated, taking into account the limitations and potential of NIRS to be used in such complex structures like soil aggregates. We concluded that the association between NIR signals of soil aggregates collected in the field and the PLSR models obtained with the NIR signals of casts produced under lab conditions is not the result of random variations, but that a pattern exists. Due to the enormous complexity of soil processes other approaches like Bayesian analysis should be explored.

## Supporting information

**S1 Table. Soil macrofauna abundance and biomass in the soil cores taken in the subalpine meadow of Central Pyrenees.**
(DOCX)

**S2 Table. Results of the calibration model performance for each species and the variables measured in incubated earthworm casts (all dates) as assessed by the coefficient of determination of calibration (R2c), the Root Mean Square Error of Cross Validation (RMSECV), the number of factors of the PLS model.** Analysis was performed with short wave NIR spectra (700 to 1,100 nm) after Savitzky-Golay 1$^{st}$ derivative transformation with 21 smoothing points (10, 10).
(DOCX)

**S3 Table. Results of the calibration model performance for each species and the variables measured in incubated earthworm casts (all dates) as assessed by the coefficient of determination of calibration (R2c), the Root Mean Square Error of Cross Validation**

**(RMSECV), the number of factors of the PLS model.** Analysis was performed with NIR spectra from 1,100 to 2,500 nm after Savitzky-Golay 2$^{nd}$ derivative transformation with 21 smoothing points (10, 10).
(DOCX)

**S1 Fig. Cross-validation of the short-wave NIRS (700–1,100 nm) calibration models in predicting C, N, $NH_4^+$ and $NO_3^-$ concentrations of casts produced by *A. rosea* in laboratory conditions.** The C:N ratio is not shown (the correlation coefficient was very low).
(PDF)

**S2 Fig. Cross-validation of the short-wave NIRS (700–1,100 nm) calibration models in predicting C, N, $NH_4^+$, $NO_3^-$ concentrations and C:N ratio of casts produced by *L. friendi* in laboratory conditions.**
(PDF)

**S3 Fig. Cross-validation of the short-wave NIRS (700–1,100 nm) calibration models in predicting C, N, $NH_4^+$, $NO_3^-$ concentrations and C:N ratio of casts produced by *P. pyrenaicus* in laboratory conditions.**
(PDF)

**S4 Fig. Cross-validation of the long-wave NIRS (1,100–2,500 nm) calibration models in predicting C, N, $NH_4^+$, $NO_3^-$ concentrations and C:N ratio of casts produced by *A. rosea* in laboratory conditions.**
(PDF)

**S5 Fig. Cross-validation of the long-wave NIRS (1,100–2,500 nm) calibration models in predicting C, N, $NH_4^+$, $NO_3^-$ concentrations and C:N ratio of casts produced by *L. friendi* in laboratory conditions.** N and NO3 concentrations are not shown (very low coefficient of correlation).
(PDF)

**S6 Fig. Cross-validation of the long-wave NIRS (1,100–2,500 nm) calibration models in predicting C, N, $NH_4^+$, $NO_3^-$ concentrations and C:N ratio of casts produced by *P. pyrenaicus* in laboratory conditions.**
(PDF)

## Acknowledgments

Field work and lab incubation was performed by all coauthors, except AZ. We are very grateful to lab technicians for their assistance in lab analyses. The ARAID foundation is acknowledged for support to Juan J. Jiménez. We acknowledge support of the publication fee by the CSIC Open Access Publication Support Initiative through its Unit of Information Resources for Research (URICI).

## Author Contributions

**Conceptualization:** Patrick Lavelle, Juan J. Jiménez.

**Data curation:** Silvia Gutiérrez-Eisman.

**Formal analysis:** Elena Velásquez, Anne Zangerlé, Patrick Lavelle, Juan J. Jiménez.

**Investigation:** Patrick Lavelle, Juan J. Jiménez.

**Methodology:** Yamileth Domínguez-Haydar, Elena Velásquez, Anne Zangerlé, Patrick Lavelle, Silvia Gutiérrez-Eisman, Juan J. Jiménez.

**Validation:** Juan J. Jiménez.

**Writing – original draft:** Juan J. Jiménez.

**Writing – review & editing:** Patrick Lavelle, Juan J. Jiménez.

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
