## [Decision Letter · Decision Letter 0]

15 May 2020

PONE-D-20-10139

Unveiling the age and origin of biogenic aggregates produced by earthworm species with their NIRS fingerprint in a subalpine meadow of Central Pyrenees

PLOS ONE

Dear Dr Jimenez,

Thank you for submitting your manuscript to PLOS ONE. After careful consideration, we feel that it has merit but does not fully meet PLOS ONE’s publication criteria as it currently stands. Therefore, we invite you to submit a revised version of the manuscript that addresses the points raised during the review process.

We would appreciate receiving your revised manuscript by recent 8 weeks. To enhance the reproducibility of your results, we recommend that if applicable you deposit your laboratory protocols in protocols.io, where a protocol can be assigned its own identifier (DOI) such that it can be cited independently in the future. For instructions see: http://journals.plos.org/plosone/s/submission-guidelines#loc-laboratory-protocols

We look forward to receiving your revised manuscript.

Kind regards,

Fuzhong Wu

Academic Editor

PLOS ONE

Additional Editor Comments:

One reviewer suggested to reject it, but I think an chance should be given to the authors because another reviewer showed positive comments. Anyway, it has highly possibility to decline it, if the authors have not addressed all of the comments.

Journal Requirements:

3. During our internal checks, the in-house editorial staff noted that you conducted research or obtained samples in another country. Please check the relevant national regulations and laws applying to foreign researchers and state whether you obtained the required permits and approvals. Please address this in your ethics statement in both the manuscript and submission information.

'..The ARAID foundation is acknowledged for support to Juan J. Jiménez. We acknowledge support of the publication fee by the CSIC Open Access Publication Support Initiative through its Unit of Information Resources for Research (URICI).'

'YDH was supported by a COLCIENCIAS grant, Colombia (Code: 1116-569-34827) for a stay at IPE-CSIC (Spain). EV, PL, AZ, SG did not receive specific funding. The funders had no role in study design, data collection and analysis, decision to publish, or preparation of the manuscript.'

5. Please include captions for your Supporting Information files at the end of your manuscript, and update any in-text citations to match accordingly. Please see our Supporting Information guidelines for more information: http://journals.plos.org/plosone/s/supporting-information

Reviewers' comments:

Reviewer's Responses to Questions

**Comments to the Author**

1. Is the manuscript technically sound, and do the data support the conclusions?

Reviewer #1: Partly

Reviewer #2: Yes

2. Has the statistical analysis been performed appropriately and rigorously? 

Reviewer #1: N/A

Reviewer #2: Yes

3. Have the authors made all data underlying the findings in their manuscript fully available?

Reviewer #1: Yes

Reviewer #2: Yes

4. Is the manuscript presented in an intelligible fashion and written in standard English?

Reviewer #1: Yes

Reviewer #2: Yes

5. Review Comments to the Author

Reviewer #1: My major concerns are:

First, the Introduction section was not written properly and can be improved by 1) giving the background information and importance of this topic to the reader; 2) identifying what are unknown and what are the questions; and 3) letting the reader know what you intend to do in order to answer these questions.

By the way, your hypothesis should come before your aims rather than the other way around.

Second, the results section was too lengthy. I even found a mass of the description about the supplementary materials. However, compared to the abundant content, the discussion section was too shallow.

Third, and most important, the experiment consists of multiple models, it is unclear how these (such as the lines269-277, vary in the “n” in the table 4) has been incorporated in the statistical models. Specifically, the post-hoc tests are not described and explained in the Methods section. Together, this makes it impossible to assess the robustness of your models, and hence, the robustness of your conclusions.

Accordingly, although the key findings of the paper are of potential interest to PLoS ONE, there are some problems still need to be addressed before it’s publication. After revision, a re-submitted version may be further considered.

Minor comments:

Line31: Please check the usage of parentheses for the first and second species.

The purposes, as well as the results, in the abstract section was too separated, such as the lines 38-41 and lines 45-47.

Line101: Please make sure that the soils all be named using WRB classification.

Line109-111: Repeat information: “dominated by…” and “as dominant plant species”

Line113: see my comments on line31. In addition, the abbreviations should be clear, you may consider to write the full names in the first mentioning in the main text.

The language needs to be improved by a native speaker before re-submission. For example, Lines129-130, and other places throughout the manuscript.

Line257-260: According to the Table 3, I can’t address the meaning the author wants to presented. Is there something error? maybe I am mistaken, but should be clarified.

Lines264-266: I couldn’t find the relevant information in the table 3 at all.

Lines273-274: see my major comments.

Lines323-330: If the relevant information is valuable. Why you put them in the supplementary material.

Lines471-472, lines 496-497 and lines 522-523: Please rephrase.

Reviewer #2: Review Domínguez-Haydar et al: Unveiling the age and origin of biogenic aggregates produced by earthworm species with their NIRS fingerprint in a subalpine meadow of Central Pyrenees

The manuscript PONE-20-10139 by Domínguez-Haydar et al presents a very interesting study aiming to acquire data on the production rate and spatio-temporal dynamics of earthworm induced macroaggregation in mountain soils and explore the capacity of NIRS to discriminate the age and origin (species) of a given soil macroaggregate (ped) under field conditions in subalpine pastures.

By combining the NIR spectral signatures of biogenic aggregates, root-aggregates, and non-aggregated soil and soil carbon (C), nitrogen (N), 4+ and 3− data, the authors suggest that a NIRS biogenic background noise (BBN) is present in the soil as a result of earthworm activity.

Overall, I think the topic of the manuscript is interesting and experiment design is rather reasonable. The results provide insights on how to analyse the role of soil fauna (e.g. earthworm) in important ecological processes of soil macroaggregation and associated organic matter dynamics by means of analyzing the BBN in the soil matrix.

However, before accepting to publication some minor revisions should be considered.

First, the results section is very detail. However, I feel that the results section is currently not well structured and the main results of the study are easy to miss in the large number of index, graphs, tables and analysis. I would recommend for the authors to thoroughly revise this section focusing more on the novel core results.

Second, the structure of the manuscript and tables should follow the guideline of the PONE.

6. PLOS authors have the option to publish the peer review history of their article (what does this mean?). If published, this will include your full peer review and any attached files.

Reviewer #1: No

Reviewer #2: No

---

## [Author Response · Author response to Decision Letter 0]

15 Jun 2020

Additional Editor Comments:

One reviewer suggested to reject it, but I think a chance should be given to the authors because another reviewer showed positive comments. Anyway, it has highly possibility to decline it, if the authors have not addressed all of the comments.

[OK, we do really thank the Editor for this decision and the chance given to us to submit a revised version of this research]

Journal Requirements:

[OK, we initially followed the guidelines of formatting and style, but it seems something went wrong. We have gone through the style requirements again, hoping that now the requirements are met.]

[OK, the site where the research was conducted is an area that has been studied by different researchers from IPE-CSIC during the last 60 years. These studies covered aspects of general ecology, botany, geomorphology and other scientific disciplines, ranging from pastoral use, plant vegetation composition and dynamics, snow dynamics, tree line succession and woody encroachment processes due to land abandonment. We are granted permission from the Government of Aragón, no paperwork required. We just need to ask permission from the mayor of the near village (Fanlo) to access the site through a restricted unpaved way. We have mentioned it in the acknowledgement section.]

3. During our internal checks, the in-house editorial staff noted that you conducted research or obtained samples in another country. Please check the relevant national regulations and laws applying to foreign researchers and state whether you obtained the required permits and approvals. Please address this in your ethics statement in both the manuscript and submission information.

[No, we have neither conducted research in another country, nor obtained samples outside Spain. The study was conducted in the Central Pyrenees, the mountainous region that limits the border between Spain and France, but all samples were collected on the Spanish side. We are attaching a map (Google Earth) to indicate the site area (see below). The yellow line sets the border between Spain and France.

'..The ARAID foundation is acknowledged for support to Juan J. Jiménez. We acknowledge support of the publication fee by the CSIC Open Access Publication Support Initiative through its Unit of Information Resources for Research (URICI).'

[OK, the sentence in the acknowledgement section “The ARAID foundation is acknowledged for support to Juan J. Jiménez” does not refer to any funding support, that is why it was not mentioned in the Funding Statement section of the online submission form. It is just that those researchers from ARAID must state that sentence in the acknowledgements section. 

With regards to the CSIC Open Access Publication Support Initiative we also need to include it in the acknowledgement section, as they will cover part of the Article Processing Charges.]

'YDH was supported by a COLCIENCIAS grant, Colombia (Code: 1116-569-34827) for a stay at IPE-CSIC (Spain). EV, PL, AZ, SG did not receive specific funding. The funders had no role in study design, data collection and analysis, decision to publish, or preparation of the manuscript.'

[Yes, this is correct]

a. Please clarify the sources of funding (financial or material support) for your study. List the grants or organizations that supported your study, including funding received from your institution.

[No salary was received]

d. If you did not receive any funding for this study, please state: “The authors received no specific funding for this work.”

[OK, we have stated this: “The authors received no specific funding for this work.”]

[OK, we have included all these statements in the rebuttal cover letter.]

5. Please include captions for your Supporting Information files at the end of your manuscript, and update any in-text citations to match accordingly. Please see our Supporting Information guidelines for more information: http://journals.plos.org/plosone/s/supporting-information

[OK, captions for supporting Information files have been included at the end of the manuscript. The figures in the Supporting Information section have been saved in pdf format, independently.]

Reviewers' comments:

Reviewer's Responses to Questions

Comments to the Author

1. Is the manuscript technically sound, and do the data support the conclusions?

Reviewer #1: Partly

Reviewer #2: Yes

2. Has the statistical analysis been performed appropriately and rigorously? 

Reviewer #1: N/A

Reviewer #2: Yes

3. Have the authors made all data underlying the findings in their manuscript fully available?

Reviewer #1: Yes

Reviewer #2: Yes

4. Is the manuscript presented in an intelligible fashion and written in standard English?

Reviewer #1: Yes

Reviewer #2: Yes

5. Review Comments to the Author

Reviewer #1: My major concerns are:

First, the Introduction section was not written properly and can be improved by 1) giving the background information and importance of this topic to the reader; 2) identifying what are unknown and what are the questions; and 3) letting the reader know what you intend to do in order to answer these questions.

[OK, we have rewritten some sections of the introduction to add clarity, emphasizing what remains unknown, the hypothesis tested in this study and the questions pending to be answered. What we intended to do in order to answer these questions was explicitly stated in the methods section]

By the way, your hypothesis should come before your aims rather than the other way around.

[OK, we have include the hypothesis before the aims of the study.]

Second, the results section was too lengthy. I even found a mass of the description about the supplementary materials. However, compared to the abundant content, the discussion section was too shallow.

[OK, we thank this reviewer comment, but since the technique is quite specific we believed it should be explained at its maximal length to ease understanding of the readers and likely researchers willing to repeat this kind of experimentation. However, in section Calibration model performance and PLSR, tables 5 and 6 have been moved to supplementary material as S2 Table and S3 table.]

Third, and most important, the experiment consists of multiple models, it is unclear how these (such as the lines269-277, vary in the “n” in the table 4) has been incorporated in the statistical models. Specifically, the post-hoc tests are not described and explained in the Methods section. Together, this makes it impossible to assess the robustness of your models, and hence, the robustness of your conclusions.

[OK, regarding data of table 4 and the use of posthoc comparisons as indicated by the reviewer #1, the analysis of variance showed that there were no significant differences in the concentration of the three elements analysed (C, N and C:N), so there is no reason to include in the table the post-hoc comparisons. We have reworded the sentence. We do hope it is clearer now.

Moreover, there are several statistical analyses in this study but this one specifically was analysed with ANOVA, and no subsequent analyses were needed. For the rest of statistical procedures, as PCA and PLS regression, these are explained in the methods section.]

Accordingly, although the key findings of the paper are of potential interest to PLoS ONE, there are some problems still need to be addressed before it’s publication. After revision, a re-submitted version may be further considered.

[OK, we do really thank the reviewer for his/her comments and for the opportunity given to resubmit a revised version of the manuscript.] 

Minor comments:

Line31: Please check the usage of parentheses for the first and second species.

[OK, only the authorship for the first species was within parentheses]

The purposes, as well as the results, in the abstract section was too separated, such as the lines 38-41 and lines 45-47.

[OK, the sentence between lines 45-47 was moved immediately after the sentence in lines 38-41]

Line101: Please make sure that the soils all be named using WRB classification.

[OK, soils have been named using WRB classification, and the reference has been updated]

Line109-111: Repeat information: “dominated by…” and “as dominant plant species”

[OK, the word dominated has been replaced by characterized and “as dominant plant species” has been kept]

Line113: see my comments on line31. In addition, the abbreviations should be clear, you may consider to write the full names in the first mentioning in the main text.

[OK, the usage of parentheses has been checked and acronym was used after first mentioning the words organic matter]

The language needs to be improved by a native speaker before re-submission. For example, Lines129-130, and other places throughout the manuscript.

[OK, the language was checked by a native English speaking person. The term at a naked eye is quite common in our discipline when describing how different structures and soil organisms found in the soil sample are hand-sorted. However we have removed the term at the naked eye.]

Line257-260: According to the Table 3, I can’t address the meaning the author wants to presented. Is there something error? maybe I am mistaken, but should be clarified.

[OK, we have rewritten this paragraph. The protocol of collecting the items in the 9 soil cores was described previously in the materials and methods section.

Now the sentence reads: “The biomass of aggregates attached to roots and non-macroaggregated soil was 122.3 and 134.8 g m-2, respectively, while biomass of free (particulate) organic matter and invertebrate biogenic aggregates was 62.9 and 41.7 g m-2, respectively (Table 3)”.]

Lines264-266: I couldn’t find the relevant information in the table 3 at all.

[The figures indicated in those lines refers to an estimation of the biomass in terms of the surface area occupied by the soil cores; each soil core had a surface area of 10×10 cm2, if we extrapolate the value of 41.7 g in 10 cm2 to Kg m-2 we obtain 41.7 Kg. We have rephrased this sentence to improve clarity].

Lines273-274: see my major comments.

[OK, regarding data of table 4 and the use of posthoc comparisons as indicated by the reviewer #1, the analysis of variance showed that there were no significant differences in the concentration of the three elements analysed (C, N and C:N), so there is no reason to include in the table the post-hoc comparisons. We have reworded the sentence. We do hope it is clearer now].

Lines323-330: If the relevant information is valuable. Why you put them in the supplementary material.

[OK, yes, thanks for this comment; we had actually included this information in the main text before submission but later was moved to supplementary information section. We have put back the figure (new figure 4) that was in the Supporting Information as S1 a-f in the original submission.]

Lines471-472, lines 496-497 and lines 522-523: Please rephrase.

[OK, the sentences have been rephrased:

Lines 471-472: Comparison of our results with those obtained in other studies is not possible as the approach used in this study, i.e., linking field aggregates of the soil matrix, not only fresh casts, to NIR signals was not assessed in previous works.

Lines 496-497: These authors collected fresh casts from the field to compare and match their NIR signals with the casts produced by the same species in the lab. Our study, on the contrary, is the first one aiming at pairing the NIRS spectra fingerprints of ageing biogenic aggregates with the respective species that likely produced those aggregates.

Lines 522-523: Octolasion lacteum and O. cyaneum were not found at the time of sampling in the field, so no casts were available for both species.]

Reviewer #2: Review Domínguez-Haydar et al: Unveiling the age and origin of biogenic aggregates produced by earthworm species with their NIRS fingerprint in a subalpine meadow of Central Pyrenees

The manuscript PONE-20-10139 by Domínguez-Haydar et al presents a very interesting study aiming to acquire data on the production rate and spatio-temporal dynamics of earthworm induced macroaggregation in mountain soils and explore the capacity of NIRS to discriminate the age and origin (species) of a given soil macroaggregate (ped) under field conditions in subalpine pastures.

By combining the NIR spectral signatures of biogenic aggregates, root-aggregates, and non-aggregated soil and soil carbon (C), nitrogen (N), 4+ and 3− data, the authors suggest that a NIRS biogenic background noise (BBN) is present in the soil as a result of earthworm activity.

Overall, I think the topic of the manuscript is interesting and experiment design is rather reasonable. The results provide insights on how to analyse the role of soil fauna (e.g. earthworm) in important ecological processes of soil macroaggregation and associated organic matter dynamics by means of analyzing the BBN in the soil matrix.

However, before accepting to publication some minor revisions should be considered.

First, the results section is very detail. However, I feel that the results section is currently not well structured and the main results of the study are easy to miss in the large number of index, graphs, tables and analysis. I would recommend for the authors to thoroughly revise this section focusing more on the novel core results.

[OK, we have tried to counterbalance the comments of both reviewers, as Reviewer 1 suggested to include one more graph in the main text. We have moved some of the tables from this section to supplementary material section and focused more on the main aspects of the analysis to explore the capacity of NIRS to discriminate the age and origin (species) of a given soil macroaggregate and emphasize the novel core results.

We also believe that it is necessary to show with some details the different steps of the analysis of the NIR spectra.].

Second, the structure of the manuscript and tables should follow the guideline of the PONE.

6. PLOS authors have the option to publish the peer review history of their article (what does this mean?). If published, this will include your full peer review and any attached files.

[OK, we agree to the option to publish the peer review history of this paper if finally accepted].

Do you want your identity to be public for this peer review? For information about this choice, including consent withdrawal, please see our Privacy Policy.

Reviewer #1: No

Reviewer #2: No

---

## [Decision Letter · Decision Letter 1]

6 Jul 2020

PONE-D-20-10139R1

Unveiling the age and origin of biogenic aggregates produced by earthworm species with their NIRS fingerprint in a subalpine meadow of Central Pyrenees

PLOS ONE

Dear Dr. Jimenez,

Thank you for submitting your manuscript to PLOS ONE. After careful consideration, we feel that it has merit but does not fully meet PLOS ONE’s publication criteria as it currently stands. Therefore, we invite you to submit a revised version of the manuscript that addresses the points raised during the review process.

We look forward to receiving your revised manuscript.

Kind regards,

Fuzhong Wu

Academic Editor

PLOS ONE

Reviewers' comments:

Reviewer's Responses to Questions

**Comments to the Author**

1. If the authors have adequately addressed your comments raised in a previous round of review and you feel that this manuscript is now acceptable for publication, you may indicate that here to bypass the “Comments to the Author” section, enter your conflict of interest statement in the “Confidential to Editor” section, and submit your "Accept" recommendation.

Reviewer #1: (No Response)

Reviewer #2: All comments have been addressed

2. Is the manuscript technically sound, and do the data support the conclusions?

Reviewer #1: Yes

Reviewer #2: Yes

3. Has the statistical analysis been performed appropriately and rigorously? 

Reviewer #1: Yes

Reviewer #2: Yes

4. Have the authors made all data underlying the findings in their manuscript fully available?

Reviewer #1: Yes

Reviewer #2: Yes

5. Is the manuscript presented in an intelligible fashion and written in standard English?

Reviewer #1: Yes

Reviewer #2: Yes

6. Review Comments to the Author

Reviewer #1: Abstract：

Redundant information: produced in the lab. produced in the lab.

In the abstract: provide more quantitative data.

Introduction：

The results will be useful for unveiling the contribution of earthworms in soil aggregation and soil organic matter (SOM) dynamics, so that this methodology can be used in other studies with other earthworm species. Rewrite the objective.

Results：

“is listed in The concentrations of C, N and C:N ratio in the three types of soil aggregates identified (Table 4)” rewrite.

Rewrite the following words or sentence: If we; We also analysed. These phrases are useless, start with the results and conclusions directly.

Discussion：

Several factors affected the variability of NIR spectral signals that were not addressed in this study:

1) Abiotic, such as water flow and dry-rewetting cycles that may disrupt aggregates, and

2) Biotic, like roots morphology, microbial biomass, and particulate organic matter in the soil.

3) human-induced, related to the process of identifying and separating BA from the soil block that may result in less precise selection.

- Delete this “conclusion”, this means that your study does not have any additional value for the research topic.

Reviewer #2: This paper is much improved after revision and it meets the standards of journal publication. I would like to "Accept"

7. PLOS authors have the option to publish the peer review history of their article (what does this mean?). If published, this will include your full peer review and any attached files.

Reviewer #1: No

Reviewer #2: No

---

## [Author Response · Author response to Decision Letter 1]

8 Jul 2020

PONE-D-20-10139R1

Unveiling the age and origin of biogenic aggregates produced by earthworm species with their NIRS fingerprint in a subalpine meadow of Central Pyrenees

PLOS ONE

Review Comments to the Author

Reviewer #1: Abstract：

Redundant information: produced in the lab. produced in the lab [OK, corrected]

In the abstract: provide more quantitative data. [OK, more quantitative data have been provided in the abstract]

Introduction：

The results will be useful for unveiling the contribution of earthworms in soil aggregation and soil organic matter (SOM) dynamics, so that this methodology can be used in other studies with other earthworm species. Rewrite the objective. [OK, the objective has been rewritten; we have changed this paragraph a bit. The main objective was to unveil the contribution of earthworms in soil aggregation and soil organic matter (SOM) dynamics in mountain soils.

The last sentence in the introduction emphasizes the need for replicated studies with other earthworm communities and/or soil invertebrates producing biogenic structures by using the same NIRS and PLS regression methodology.]

Results：

“is listed in The concentrations of C, N and C:N ratio in the three types of soil aggregates identified (Table 4)” rewrite. [OK, corrected. “The concentrations of C, N and C:N ratio in the three types of soil aggregates identified (Table 4) did not differ statistically (ANOVA, p>0.05)]

Rewrite the following words or sentence: If we; We also analysed. These phrases are useless, start with the results and conclusions directly. [OK, the sentences have been rewritten. “When these data were extrapolated to kg per meter square, the activity of …..”]

Discussion：

Several factors affected the variability of NIR spectral signals that were not addressed in this study:

1) Abiotic, such as water flow and dry-rewetting cycles that may disrupt aggregates, and

2) Biotic, like roots morphology, microbial biomass, and particulate organic matter in the soil.

3) human-induced, related to the process of identifying and separating BA from the soil block that may result in less precise selection.

- Delete this “conclusion”, this means that your study does not have any additional value for the research topic. [OK, these factors were not tested in this study. We have removed all factors. We have however included a sentence about the importance of expertise of students, researchers, etc. in the identification and selection of biogenic aggregates from field samples.]

Reviewer #2: This paper is much improved after revision and it meets the standards of journal publication. I would like to "Accept"

[OK, we are very grateful to the reviewer for the acceptance of this paper]

---

## [Decision Letter · Decision Letter 2]

21 Jul 2020

Unveiling the age and origin of biogenic aggregates produced by earthworm species with their NIRS fingerprint in a subalpine meadow of Central Pyrenees

PONE-D-20-10139R2

Dear Dr. Jimenez,

We’re pleased to inform you that your manuscript has been judged scientifically suitable for publication and will be formally accepted for publication once it meets all outstanding technical requirements.

Kind regards,

Fuzhong Wu

Academic Editor

PLOS ONE

Additional Editor Comments (optional):

Reviewers' comments:

Reviewer's Responses to Questions

**Comments to the Author**

1. If the authors have adequately addressed your comments raised in a previous round of review and you feel that this manuscript is now acceptable for publication, you may indicate that here to bypass the “Comments to the Author” section, enter your conflict of interest statement in the “Confidential to Editor” section, and submit your "Accept" recommendation.

Reviewer #1: All comments have been addressed

2. Is the manuscript technically sound, and do the data support the conclusions?

Reviewer #1: Yes

3. Has the statistical analysis been performed appropriately and rigorously? 

Reviewer #1: Yes

4. Have the authors made all data underlying the findings in their manuscript fully available?

Reviewer #1: Yes

5. Is the manuscript presented in an intelligible fashion and written in standard English?

Reviewer #1: Yes

6. Review Comments to the Author

Reviewer #1: (No Response)

7. PLOS authors have the option to publish the peer review history of their article (what does this mean?). If published, this will include your full peer review and any attached files.

Reviewer #1: No

---

## [Editor Report · Acceptance letter]

24 Jul 2020

PONE-D-20-10139R2 

Unveiling the age and origin of biogenic aggregates produced by earthworm species with their NIRS fingerprint in a subalpine meadow of Central Pyrenees 

Dear Dr. Jiménez:

I'm pleased to inform you that your manuscript has been deemed suitable for publication in PLOS ONE. Congratulations! Your manuscript is now with our production department. 

Kind regards, 

on behalf of

Professor Fuzhong Wu 

Academic Editor

PLOS ONE